# A Critical Review of Chloroquine and Hydroxychloroquine as Potential Adjuvant Agents for Treating People with Cancer

**Amal Kamal Abdel-Aziz** [1,2,3,*] **, Mona Kamal Saadeldin** [1,4] **, Ahmed Hamed Salem** [5,6] **, Safaa A. Ibrahim** [7,8] **, Samia Shouman** [9] **, Ashraf B. Abdel-Naim** [10] **and Roberto Orecchia** [11]

1   Department of Experimental Oncology, IEO, European Institute of Oncology IRCCS, 20139 Milan, Italy
2   Department of Pharmacology and Toxicology, Faculty of Pharmacy, Ain Shams University, Cairo 11566, Egypt
3   KAUST Smart-Health Initiative and Biological and Environmental Science and Engineering (BESE) Division, King Abdullah University of Science and Technology (KAUST), Jeddah 23955, Saudi Arabia
4   Electrical Engineering and Biological Sciences Departments, University of Notre Dame, Notre Dame, IN 46556, USA
5   Department of Experimental and Clinical Pharmacology, University of Minnesota, Minneapolis, MN 55455, USA
6   Department of Clinical Pharmacy, Faculty of Pharmacy, Ain Shams University, Cairo 11566, Egypt
7   Department of Microbiology and Immunology, Chicago Medical School, North Chicago, IL 60064, USA
8   Department of Microbiology and Immunology, Faculty of Pharmacy, Cairo University, Cairo 11562, Egypt
9   Cancer Biology Department, National Cancer Institute, Cairo University, Cairo 11796, Egypt
10  Department of Pharmacology and Toxicology, Faculty of Pharmacy, King Abdulaziz University, Jeddah 21589, Saudi Arabia
11  Scientific Directorate, IEO, European Institute of Oncology, IRCCS, 20141 Milan, Italy
*   Correspondence: amalabdel-aziz@pharma.asu.edu.eg; Tel.: +39-3881532836

**Abstract:** Chloroquine (CQ) and hydroxychloroquine (HCQ) have been used to treat malaria and autoimmune diseases for more than 70 years; they also have immunomodulatory and anticancer effects, which are linked to autophagy and autophagy-independent mechanisms. Herein, we review the pharmacokinetics, preclinical studies and clinical trials investigating the use of CQ and HCQ as adjuvant agents in cancer therapy. We also discuss their safety profile, drug–drug and drug–disease interactions. Systematic studies are required to define the use of CQ/HCQ and/or their analogues in cancer treatment and to identify predictive biomarkers of responder subpopulations.

**Keywords:** autophagy; chloroquine; hydroxychloroquine; cancer; pharmacokinetic; toxicity; repositioning

## 1. Introduction

In 1930, quinacrine (an acridine derivative) was introduced for treating malaria [1]. Its toxicity and limited efficacy stimulated the synthesis of chloroquine (CQ) in which the acridine ring of quinacrine was replaced with a quinoline ring [2]. Repurposing the use of CQ for treating patients with autoimmune diseases stemmed—at least partially—from observations during World War II that cutaneous rashes and arthritis improved in soldiers who received CQ and quinacrine as prophylaxis against malaria [2]. Hydroxychloroquine (HCQ) was synthesized later and, owing to its favorable safety profile compared to CQ [3], HCQ has been used for decades in the treatment of autoimmune diseases as systemic lupus erythematosus (SLE) and rheumatoid arthritis (RA) [2,4].

There is at least preclinical evidence that CQ and HCQ have anticancer activity. Below, we discuss their pharmacokinetics and the preclinical studies and clinical trials investigating the use of CQ or HCQ in cancer therapy.

## 2. Pharmacokinetics

Chloroquine and HCQ are well-absorbed orally, reaching peak plasma levels within 2–4.5 h [5–7]. In plasma, 30–40% are bound to albumin and $\alpha_1$-acid glycoprotein [6].Their stereoisomers exhibit differential binding, metabolism and activity [2,6].

Chloroquine and HCQ can bind to melanin in pigmented tissues, mononuclear cells and muscles [2,6]. During prolonged treatment, they accumulate with higher concentration in the heart, liver, brain, muscle and skin than in blood and their tissue concentration may correlate better with their efficacy than their blood levels [2,7].

In the liver, CQ is de-alkylated via cytochrome P450 (mainly CYP3A, CYP2C8 and CYP2D6) into the pharmacologically active desethyl CQ and bisdesethyl CQ metabolites [8,9]. HCQ is metabolized via CYP3A4 driven dealkylation into three active metabolites: desethyl CQ, desethyl HCQ and bisdesethyl HCQ [10]. Almost 40–50% of CQ and HCQ are excreted via the kidneys [6]. Being amphiphilic weak bases, CQ and HCQ are partially protonated at physiological pH (7.4) and biprotonated at acidic pH (4–5). Alkalinisation increases and acidification decreases the renal excretion of CQ [1]. CQ and HCQ have long terminal elimination half-lives (~40–50 days) [6,7].

## 3. Preclinical Studies of Anticancer Activity of CQ and HCQ

Several studies have described the anticancer potential of CQ and HCQ when used with standard cancer therapy [4,11–13]. CQ and HCQ elicit direct and indirect effects on cancer cells [12–16]. One mechanism of action is the inhibition of autophagy [2,4,11,12]. Autophagy (or self-eating) is a double-edged process, which can promote either cancer cell survival or death [17,18]. As an adaptive mechanism, some cancer cells exploit autophagy to survive during stressful conditions, such as nutrient deprivation, hypoxia or cytotoxic insults triggered by cancer therapy [11,12,19–21]. Mimicking tumor microenvironment by co-culturing cancer cells with fibroblasts promotes autophagy [22]. Conversely, the excessive induction of autophagy by diverse anticancer drugs has been reported to trigger autophagic cell death (or programmed cell death type II) of cancer cells [23–25]. CQ and HCQ act at the late stages of autophagy by raising lysosomal pH, which inhibits the fusion between autophagosomes and lysosomes, and thereby impairs lysosomal protein degradation [4,12]. Palmitoyl-protein thioesterase 1 has been identified as the lysosomal target of CQ/HCQ [26]. It is worth mentioning that the synergistic anticancer activity of CQ and temozolomide combination was abrogated with the pharmacological or genetic inhibition of early stages of autophagy [27]. Knocking down p53 or overexpressing mutant p53 also compromised the anticancer potential of CQ-temozolomide combination [27]. Notably, superior anticancer efficacy of CQ and erlotinib combination was maintained in the preclinical "cancer cells/fibroblasts co-culturing" setting, mimicking the tumor microenvironment [22]. Given the regulatory crosstalk between autophagy and apoptosis, the augmentation of the anticancer efficacy of chemotherapy by CQ or HCQ might be associated with increased apoptosis [12,14]. Indeed, the anti-apoptotic Bcl-2 family members as Bcl-2 and Bcl-xl inhibit autophagy [28]. CQ augmented the anticancer activity of Bcl-2 inhibitors [29,30]. The overexpression of Bcl-2 or Bcl-xL compromised apoptotic cancer cell death triggered in ABT-737 (Bcl-2 inhibitor), CQ and their combination [30].

The physicochemical properties of CQ and HCQ present a critical limitation that restrains their anticancer activity [31]. Solid tumours often develop an insufficient vasculature to supply their nutrient needs and contain regions of hypoxia. Hypoxic cancer cells depend on glycolysis, and other cancer cells may use glycolysis electively for ATP synthesis. This promotes an acidic extracellular microenvironments, which compromises the cellular uptake of CQ/HCQ [31]. To overcome this shortcoming, a series of CQ/HCQ derivatives has been synthesized to increase their anticancer potential [31–35]. For instance, Lys05, a dimeric CQ, has been reported to be a more potent inhibitor of autophagy with greater anticancer activity than HCQ [34,35]. However, chronic daily treatment of mice with Lys05 was associated with Paneth cell dysfunction, although without obvious signs of

gastrointestinal toxicity [35]. Since inhibitors of autophagy have been shown to improve the effects of chemotherapy in preclinical models, they should undergo clinical evaluation.

CRISPR-Cas9 loss-of-function screening identified insulin-like growth factor 1 receptor (IGF1R) as a sensitizer of pancreatic ductal adenocarcinoma cells to CQ/HCQ [36]. Co-targeting IGF1R and ERK inhibited glycolysis and augmented the dependence of pancreatic ductal adenocarcinoma cells on autophagy, and hence rendered them more vulnerable to CQ/HCQ [36].

Chloroquine has also been reported to augment the vulnerability of cancer cells to chemotherapy via autophagy-independent mechanisms, including the normalization of tumour vasculature, which decreases intratumoral hypoxia, cancer cell invasion and metastasis [37,38]. Notably, CQ-induced vessel normalization was linked to activated endothelial Notch1 signaling [39,40]. CQ also sensitized triple negative breast cancer cells to paclitaxel via reducing $CD44^+/CD24^{-/low}$ cancer stem cells [41].

Chloroquine and HCQ have been reported to inhibit angiogenesis [12]; they also induced the secretion of the tumour suppressor prostate apoptosis response-4 (Par-4) from normal cells of treated mice and cancer patients, which triggered paracrine apoptosis of cancer cells and inhibited tumour metastasis [16]. Furthermore, CQ promoted the anti-tumour immune responses via resetting tumour-associated macrophages from the M2 to the tumour-killing M1 phenotype [15]. CQ increases macrophage lysosomal pH and triggers $Ca^{2+}$ release through the lysosomal $Ca^{2+}$ channel mucolipin-1, which activates p38 and nuclear factor kappa B (NF-κB), thereby causing tumour-associated macrophages to adopt an M1 phenotype [15]. The antitumor immune responses provoked by CQ were observed preclinically when used at relatively high concentration (10 μM). However, the safe plasma level/concentration of CQ is reported to be approximately 3 μM [42] so that the antitumor immunogenic doses of CQ might not be tolerable clinically. The limited clinical efficacy of immune checkpoint inhibitors as monotherapy has instigated the investigating of its inclusion in diverse combinatorial regimens [43,44]. Of note, HCQ compromised the anticancer T cell immune responses triggered by anti-PD1 in syngeneic tumor mouse models [45]. Thus, caution is warranted, since prolonged exposure to clinically approved doses of CQ and HCQ may suppress immune responses, as occurs during their use in treating autoimmune diseases [2].

## 4. Dose and Schedule

To optimize the dose and schedule of CQ/HCQ, their target therapeutic concentrations must be identified. Three strategies might be used to achieve this objective: (i) Use of **in vitro** models to evaluate the pharmacodynamics (PD) of CQ and HCQ against cancer cells. In simple models, cancer cells are exposed to constant concentrations of the agent, but to better understand the impact of the pharmacokinetic (PK) profile on anticancer activity, dynamic **in vitro** models can be used to expose cancer cells to fluctuating concentrations of CQ and HCQ. (ii) Use of animal models, which better mimic the PK profile and immune system in people. Such models also allow dose fractionation studies to better understand the interplay between PK and PD. (iii) Use of clinical data to determine the dose and schedule that show the best correlation with the anticancer activity of these agents. This would require the use of different dosage regimens for CQ and HCQ and assessment of their plasma concentrations. In comparing CQ and HCQ, it is critical to account for the protein binding of both agents, since only free drug is pharmacologically active.

Once the therapeutic target concentrations for CQ and HCQ have been estimated, a population PK model can be used to estimate their concentrations for different doses and schedules. These models already exist for CQ and HCQ, based on data from patients with malaria, RA or SLE [46–48]. In Japanese subjects, weight was found to influence HCQ PK; therefore, weight based dosage regimens of HCQ should be considered [46]. Assuming cancer has no impact on the PK of CQ and HCQ, these models can be used to optimize the dose and schedule of CQ and HCQ in cancer patients. Coupling PK/PD models with

Monte Carlo simulation [49,50] could be utilized to compare the anticancer efficacies of different doses of CQ and HCQ.

## 5. Safety Profile

Chloroquine and HCQ have some adverse effects, associated especially with their long-term use [5,51]. Cardiac disorders have been reported in patients treated for a median of 7 years [range:3 days–35 years] with a high cumulative dose (median 800 g CQ or 1235 g HCQ) [51]. Among the cardiotoxic effects associated with their use is bundle or atrio-ventricular block, prolonged QT interval and Torsade de Pointes (TdP) [5,51]. The risk factors for developing TdP include female gender, age (>65 years), history of drug-induced TdP, chronic renal or hepatic insufficiency, electrolyte abnormalities, diuretics and simultaneous use of QT-prolonging drugs [52]. Approximately half of the patients who discontinued treatment recovered normal heart function, but the remaining patients either suffered from irreversible cardiac damage or died (24 of 127) [51]. For cancer treatment, the drugs would usually be administered for a shorter period, typically one year or less, but patients should be checked for cardiac toxicity and treatment should be withdrawn if cardiac manifestations are present.

Retinopathy is associated more frequently with CQ than HCQ [3,5,51]. CQ and HCQ bind to melanin and inhibit the lysosomal activity in the retinal pigment epithelium (RPE), and hence reduce the phagocytosis of shed photoreceptor outer segments (shed rod and cone debris), resulting in their accumulation and damage of the macular cones outside of the fovea. RPE cells, thus, migrate into the outer nuclear and plexiform layers of the retina, resulting in irreversible photoreceptor loss and RPE atrophy [53]. A retrospective case series of patients with NSCLC who received HCQ (1000 mg/day) together with erlotinib reported that two of seven patients who had been treated for ≥6 months developed retinal toxicity—without symptomatic visual acuity loss—at 11 and 17 months of exposure. Although fundus autofluorescence imaging was normal, the retinal damage was identified by optical coherence tomography and multifocal electroretinography testing [54]. Thus, long-term use of HCQ (1000 mg/day) may incite retinal toxicity within 1–2 years and sensitive retinal screening tests are needed [54].

Besides its role in cancer, autophagy plays a homeostatic role in normal cells, including heart, kidney and liver [55]. Preclinical studies have demonstrated that autophagy protects against cisplatin-induced acute nephrotoxicity and inhibiting autophagy using CQ exacerbates cisplatin-induced acute kidney damage [55]. Thus, monitoring the function of vital organs during therapy is essential.

Although the anti-malarial doses of CQ and HCQ are generally considered safe during pregnancy and breastfeeding, the safety of long-term use of higher doses for treating SLE and RA in pregnant or breastfeeding women is controversial [5,56,57].

Acute CQ poisoning (oral doses ≥ 50 mg/kg) can be lethal [5]. Intoxicated patients present with nausea and vomiting followed by slurred speech, agitation, breathlessness owing to pulmonary oedema, convulsions, arrhythmia and coma [5]. Quinidine-like cardiotoxicity has been reported following acute CQ poisoning. CQ blocks the rapid component of the delayed rectifying outward potassium current I, sodium and calcium channels, which leads to membrane-stabilization effects (resulting in AV block, QRS interval widening and QT prolongation), negative inotropic effects and peripheral vasodilatation [51,58]. The management of intoxicated cases within the first hours of CQ ingestion comprises prevention of further absorption. Otherwise, symptomatic treatment is required to maintain cardiac and respiratory functions. Diazepam can be used to control convulsions [5].

## 6. Clinical Trials

Several Phase I/II trials have evaluated the safety and efficacy of CQ and HCQ as monotherapy or in combination with surgery, radiotherapy or chemotherapy in treating patients with solid and hematological tumours (Table 1) [2,59–67]. Long-term treatment with HCQ (600 mg BID: the highest FDA-recommended dose) appears to be well-tolerated

when given with anticancer therapy [61,66–69]. CQ and HCQ have negligible anticancer efficacy when used alone [60,70], but their long-term use in pre- and post-operative cancer patients has been associated with favorable clinical outcomes [61,64,65].

A meta-analysis of seven trials evaluating the addition of CQ or HCQ to standard cancer therapy (chemotherapy or radiation) in different types of cancer (glioblastoma, brain metastases from non-small cell lung cancer (NSCLC) and breast cancer, non-Hodgkin lymphoma and pancreatic adenocarcinoma) concluded that their use was associated with improvements in overall response rate (ORR), progression-free survival (PFS) and overall survival (OS) [71]. Subgroup analysis revealed that CQ/HCQ-based therapy led to an improved 6-month PFS and 1-year OS in patients with glioblastoma, and to a higher ORR in patients with non-Hodgkin lymphoma. However, no significant improvement of ORR and 6-month PFS was found in patients with NSCLC or breast cancer. This meta-analysis has several limitations, since it included clinical studies that reported the effects of different treatment schedules of HCQ or CQ in different types of cancer, and did not provide information about long-term outcomes or the safety of the combination regimens [71].

A phase II randomized clinical trial (NCT01506973) has reported that adding HCQ to standard chemotherapy (gemcitabine and nab-paclitaxel) did not improve OS in patients with metastatic pancreatic adenocarcinoma [72]. Nonetheless, the ORR was significantly higher in the HCQ group, with a trend toward improved PFS, suggesting that a subpopulation of patients may benefit from the addition of HCQ. Another randomized clinical trial (NCT01978184) has revealed that preoperative addition of HCQ to gemcitabine and nab-paclitaxel resulted in better pathologic responses in patients with resectable pancreatic adenocarcinoma [61].

There are currently no sensitive and reliable predictive biomarkers that could guide clinicians towards the rational selection of cancer patients who could most likely benefit from CQ/HCQ. However, some studies suggested a handful of biomarkers in a limited number of patients, which warrant their further validation in clinical trials with larger cohorts [69,73,74]. Indeed, Fei and colleagues have retrospectively analyzed SMAD4 expression in pancreatic adenocarcinoma specimens of patients who were enrolled in two clinical trials (NCT01128296 and NCT01978184) evaluating the addition of pre-operative HCQ to neoadjuvant chemotherapy. The addition of HCQ was associated with better histopathologic response in pancreatic adenocarcinoma patients with SMAD4 loss [73]. There was a trend toward improved median OS—despite being statistically insignificant—in HCQ-treated patients with SMAD4 loss [73]. However, the results of this study should be interpreted with caution given its retrospective nature with data gathered from two clinical trials investigating different chemotherapy regimens. Prolonged disease-free survival and OS have been observed in pancreatic adenocarcinoma patients with > 51% increment in the peripheral blood levels of LC3-II—microtubule-associated proteins 1A/1B light chain 3B, which is used as an autophagic marker [69]. Conversely, p53 status did not correlate with the clinical outcome [69]. Elevated plasma levels of Par-4—but not tumor levels of sequestosome-1/p62 (which is used as a marker of inhibition of autophagic flux)—correlated with induced apoptosis in the tumor specimens of HCQ-treated patients [75].

**Table 1.** Clinical trials of chloroquine (CQ) and hydroxychloroquine (HCQ) in cancer therapy.

| Drug | Cancer Type | Treatment Schedule of CQ or HCQ (*n* = Sample Size) | Clinical Outcome | Ref |
|---|---|---|---|---|
| CQ | Glioblastoma multiforme | CQ (150 mg dose/day) was administered 24 h post-surgery and continued with radiotherapy and chemotherapy throughout the observation period (24–50 months) (*n* = 9/control cohort and *n* = 9/CQ cohort). | CQ prolonged the survival compared to the controls. | [64] |
| | Glioblastoma multiforme | CQ (*n* = 6: 200 mg, *n* = 3: 300 mg and *n* = 4: 400 mg) was started 1 week before chemoradiation (temozolomide + radiotherapy). | - MTD of CQ = 200 mg.<br>- Median survival was 11.5 and 20 months for EGFRvIII⁻ and EGFRvIII⁺ patients, respectively.<br>- Tolerability and OS supported further clinical studies. | [76] |
| | Brain metastases from solid tumours | Whole brain irradiation (30 Gy in 10 fractions over two weeks) together with CQ (150 mg/day were administered 1 h before whole brain irradiation and continued for 4 weeks) (*n* = 34/placebo cohort and *n* = 39/CQ cohort). | - CQ improved the control of brain metastasis, compared to control arm.<br>- No differences in OS, response rate, QoL or toxicity in either arm. | [65] |
| | Breast cancer | 500 mg/day as monotherapy for 2–6 weeks before surgery (*n* = 24/placebo cohort and *n* = 46/CQ cohort). | - No significant effect on breast cancer proliferation (Ki67).<br>- All AEs were grade 1, but caused ~15% to discontinue therapy. | [60] |
| | Metastatic or unresectable pancreatic cancer | 3+3 dose escalation study in which patients received single weekly dose of gemcitabine followed by single weekly doses of CQ (100, 200 or 300 mg) (*n* = 9). | - CQ addition to gemicitabine was well tolerated. | [74] |
| | Pancreatic adenocarcinoma | Pre-operative gemcitabine + HCQ (1200 mg/kg/day) for 31 days until surgery (*n* = 35). | - No dose-limiting toxicities and grade 4/5 treatment-related AEs.<br>- Gemcitabine and HCQ improved the OS, compared with a previous institutional cohort. | [69] |
| | Metastatic pancreatic cancer | Patients received (*n* = 10: 400 mg or *n* = 10: 600 mg) HCQ BID. | - At 2 months, 2 (10%) without PD.<br>- Median PFS and OS were 46.5 and 69.0 days, respectively.<br>- Tolerability and efficacy were similar in both dosing. | [70] |
| | Resectable pancreatic adenocarcinoma | Preoperative HCQ (600 mg BID) (*n* = 30/nab-paclitaxel and gemcitabine (PG) cohort and *n* = 34/HCQ + PG cohort). | - Preoperative HCQ (600 mg BID), gemcitabine and nab-paclitaxel conferred better pathological and serum biomarker responses and was associated with autophagy inhibition and increased immune cell tumour infiltration, compared to preoperative PG.<br>- No difference in serious AEs, OS and recurrence-free survival. | [61] |
| | Advanced or metastatic pancreatic adenocarcinoma | HCQ (600 mg BID) (*n* = 57/PG cohort and *n* = 55/PG + HCQ cohort). | - Addition of HCQ (600 mg BID) to gemcitabine and nab-paclitaxel did not improve OS at 12 months.<br>- HCQ significantly increased the overall response rate from 21% to 38%. | [72] |
| | Non-small cell lung cancer | - Patients were randomly assigned into either HCQ (*n* = 8) or HCQ + erlotinib (*n* = 19) cohorts.<br>- 3+3 Dose escalation study in which patients initially received 400 mg/day HCQ with 200 mg increment to reach a maximum dose of 1000 mg HCQ. | - Recommended Phase II dose: HCQ (1000 mg/day) + erlotinib.<br>- 28-day cycles continued until PD or unacceptable toxicity.<br>- No dose-limiting toxicities. | [77] |

**Table 1.** *Cont.*

| Drug | Cancer Type | Treatment Schedule of CQ or HCQ (*n* = Sample Size) | Clinical Outcome | Ref |
|------|-------------|------------------------------------------------------|------------------|-----|
| HCQ | Advanced solid tumours and melanoma | HCQ (200–1200 mg/day) + temozolomide for 7/14 days (*n* = 37). | - Well tolerated without recurrent dose-limiting toxicity.<br>- MTD was not reached for HCQ.<br>- Recommended Phase II dose: HCQ (600 mg BID)+ temozolomide.<br>- PR [3/22 (14%)] in metastatic melanoma patients. | [67] |
| | Glioblastoma multiforme | Phase I: HCQ (200 to 800 mg/day) with radiotherapy and temozolomide (*n* = 16). Phase II: HCQ (200 to 800 mg/day) with radiotherapy and temozolomide (*n* = 76). | - HCQ (MTD = 600 mg/day) with radiotherapy + temozolomide was associated with inconsistent autophagy inhibition and no improvement in OS. | [78] |
| | Advanced metastatic colorectal cancer | HCQ (600 mg/day) + vorinostat in a 3-week cycle (*n* = 20). | - 40% had Grade 3/4 treatment-related AEs: fatigue, nausea, vomiting, and anaemia.<br>- The combination was associated with boosted anti-tumour immunity and autophagy inhibition. | [63] |
| | Refractory/ relapsed myeloma | Two week run-in of HCQ as a monotherapy (100, 200, 400, 800 or 1200 mg/day) followed by combination therapy with bortezomib (*n* = 25). | - Recommended Phase 2 dose: HCQ (600 mg BID) for 56 days + bortezomib.<br>- Dose-related GIT toxicity and cytopenias were noticed.<br>- Of 22 patients, 3 (14%) had very good PR. | [66] |
| | Renal cell carcinoma | Everolimus + HCQ (400 or 600 mg BID) (*n* = 38). First Cycle (35 days): 1-week everolimus alone. Subsequent cycles (28 days/cycle): everolimus +HCQ. | - No dose-limiting toxicity in Phase I.<br>- Recommended Phase II dose: HCQ (600 mg BID) + everolimus. | [68] |
| | Early-stage solid tumors | 200 or 400 mg BID for 14 days before surgery (*n* = 9). | - Well-tolerated with no dose limiting toxicities or serious AEs.<br>- Tumors from the eight HCQ-treated patients with high plasma Par-4 levels underwent apoptosis.<br>- P62/sequestsome-1 was induced in tumors of all nine HCQ-treated patients. | [75] |
| | Chronic phase chronic myeloid leukemia | Imatinib (*n* = 30) or imatinib + HCQ (400 mg BID) (*n* = 32) for 12 months. | Imatinib + HCQ was tolerated with modest improvement in BCR-ABL1 qPCR levels at 12 and 24 months. | [59] |
| | Advanced BRAFV600-mutant melanoma | Patients (*n* = 38) were treated with dabrafenib and trametinib for one week and then HCQ (starting Phase I dose = 400 mg BID) was co-administered. Treatment continued until PD, and after PD in the case of isolated progression, which could be locally treated. | - The combination regimen was tolerable.<br>- PFS did not meet the prespecified threshold, but tended to be promising in patients with elevated LDH and prior treatment.<br>- Randomized study has been launched. | [79] |

AEs: adverse events, CQ: chloroquine, HCQ: hydroxychloroquine, MTD: maximum tolerated dose, OS: overall survival, PD; progressive disease, PG: nab-paclitaxel and gemcitabine, PR: partial response, SD: stable disease, QoL: quality of life.

## 7. Drug-Drug and Drug-Disease Interactions

Chloroquine and HCQ interact with several drugs, yet the molecular basis and magnitude/incidence for many remain unknown (Table 2). Some of these PK drug–drug interactions might be attributed to the modulatory effects of CQ/HCQ on the activity of some cytochrome P450 (CYP) metabolizing enzymes and/or p-glycoprotein [9,80–82].

**Table 2.** Known and potential drug–drug interactions of chloroquine (CQ) and hydroxychloroquine (HCQ).

| Drug | Interacting Drug | Type of Interaction/Recommendations | Ref |
|---|---|---|---|
| CQ | Some antacids | Some antacids decreases CQ bioavailability and time spacing >4 h is recommended. | [83] |
| CQ | Ampicillin | CQ decreases the bioavailability of ampicillin | [84] |
| CQ/HCQ | QT Prolongation inducing drugs | Co-administration of >1 QT prolonging drugs can increase the risk of developing prolonged QT associated-arrthymia. | [51,52,58] |
| CQ/HCQ | Tamoxifen | Increased risk for retinopathy. | [56,85] |
| CQ | Ciclosporin (cyclosporin) | Three-day CQ administration was associated with elevated serum ciclosporin and creatinine levels which was reversed one week after CQ discontinuation. | [86] |
| CQ | Methotrexate | CQ decreases the area plasma under the curve of methotrexate. | [87] |
| CQ | Cimetidine | Cimetidine impairs CQ elimination. | [88] |
| CQ | Acetaminophen (Paracetamol) | CQ increases the peak plasma levels and AUC of paracetamol. | [89] |
| CQ | Primaquine | CQ increases the plasma levels of primaquine and carboxyprimaquine and its use is associated with slight corrected QT (QTc) interval prolongation. | [90] |
| CQ | Digoxin | CQ increases the serum levels of digoxin which warrants careful monitoring. | [91] |
| CQ | Cisplatin | CQ exacerbates acute cisplatin-induced nephrotoxicity. | [55] |
| HCQ | Insulin and hypoglycaemic drugs | HCQ induces hypoglycaemia and dose re-adjustment of insulin or hypoglycaemic drugs is necessary. | [92] |
| HCQ | Metoprolol | HCQ increases the bioavailability of metoprolol. | [80] |

Pharmacodynamic drug–drug interactions of CQ/HCQ with other QT prolonging drugs could increase the risk for developing TdP. Some anticancer drugs (such as sunitinib, cabozantinib and lapatinib) are associated with QT prolongation [52] and should not be used in combination with CQ or HCQ. Nausea and vomiting, which are frequently associated with anticancer therapy, may lead to dehydration followed by electrolyte imbalance, and thus provoke QT prolongation [52]. QT prolongation by some drugs is associated with increased incidence of arrhythmic death [52]. Using CQ or HCQ alone or with QT-prolonging anticancer drugs mandates careful cardiac monitoring and correction of any electrolyte abnormalities. The management of potentially fatal arrhythmias associated with prolonged QT syndrome involves the intravenous administration of magnesium sulphate and electrical cardioversion [52].

Tamoxifen decreases the activity of cathepsin D, a lysosomal acid protease, in the lysosomes of RPE, which is essential for the phagocytosis of the ingested rod outer segments shed from photoreceptor cells [85]. Increased risk factors for retinopathy include co-treatment with tamoxifen and HCQ, >5 mg/kg/day HCQ, pre-existing maculopathy and renal insufficiency [56].

Glucose 6-phosphate dehydrogenase (G6PD) protects RBCs against oxidative stress, which is triggered by CQ [93].Thus, CQ may cause hemolysis in patients with G6PD deficiency [93]. CQ/HCQ may also increase the risk of convulsions in patients with epilepsy [94,95]. Their epileptogenic potential is linked to inhibition of GABA and the enhancement of dopaminergic neurotransmissions [94,95].

## 8. Conclusions and Future Perspectives

Critical appraisal of the challenges of using CQ and HCQ as adjuvant agents in cancer patients can be summarized as follows:

i.     Despite their preclinical anticancer and safety profile, there are currently no sensitive and reliable predictive biomarkers for the rational selection of cancer patients who

could benefit from the use of CQ/HCQ and avoid the exposure of non-responders to their adverse effects.

ii. Consideration of the risk and benefit for CQ/HCQ in cancer patients must be individualized.

iii. Some anticancer drugs and HCQ/CQ are associated with a prolonged QT interval. Given the multifactorial developmental nature of TdP, careful cardiac monitoring and correction of electrolyte imbalance are critical and treatment withdrawal is needed if cardiac manifestations arise.

iv. Given the homeostatic role of autophagy, which is inhibited by HCQ/CQ, monitoring of vital organs is essential.

v. Long-term follow-up of treated patients for potential cardiovascular, renal and retinal toxicities is warranted.

**Author Contributions:** Conceptualization, A.K.A.-A.; methodology, A.K.A.-A., M.K.S., A.H.S., S.A.I., S.S., A.B.A.-N., R.O.; data curation, A.K.A.-A.; writing—original draft preparation, M.K.S., A.H.S., S.A.I., S.S., A.B.A.-N., R.O.; writing—review and editing. All authors have read and agreed to the published version of the manuscript.

**Funding:** This research received no external funding.

**Institutional Review Board Statement:** Not applicable.

**Informed Consent Statement:** Not applicable.

**Data Availability Statement:** Not applicable.

**Acknowledgments:** The authors would like to thank Ian Tannock and Houriya Elbarbary for the constructive feedback and critical discussions.

**Conflicts of Interest:** The authors declare no conflict of interest.

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
