# Peer review of "A Critical Review of Chloroquine and Hydroxychloroquine as Potential Adjuvant Agents for Treating People with Cancer"

_futurepharmacol, doi:10.3390/futurepharmacol2040028_

Round 1

Reviewer 1 Report

Increasing evidence, including preclinical studies and ongoing clinical trials, has shown that chloroquine (CQ) and hydroxychloroquine (HCQ) exhibit promising anti-cancer effects in various cancers. The authors nicely discussed this critical topic in the review. The MS is well written and contains necessary information regarding the clinical application of CQ/HCQ for cancer treatment. The MS can be accepted after addressing the following comments:

1. Table 1 contains the most important information regarding clinical application of CQ/HCQ. The effect of CQ/HCQ on cancer treatment can be moved to a new colume with more detailed information. The patients number (sample size) of clinical trial should be provide

2. The authors raise an important question (there are currently no sensitive and reliable predictive biomarkers for rational selection of cancer patients who could benefit from using CQ/HCQ). Some studies have shown that CQ and HCQ inhibit anti-cancer therapy-induced protective autophagy (e.g., chemotherapy, radiotherapy, and target therapy) to exert anti-cancer effect. The authors may discuss the potential of using autophagy flux as the biomarker for CQ/HCQ  

3. Please leave a space between the last character and the left parenthesis (e.g., line 38), or between two sentences (e.g., line 60, 79, 91).

Author Response

Reviewer 1: Increasing evidence, including preclinical studies and ongoing clinical trials, has shown that chloroquine (CQ) and hydroxychloroquine (HCQ) exhibit promising anti-cancer effects in various cancers. The authors nicely discussed this critical topic in the review. The MS is well written and contains necessary information regarding the clinical application of CQ/HCQ for cancer treatment. The MS can be accepted after addressing the following comments:

RE: We thank the reviewer for her/his interest in our review and the additional points made, that allowed us to better formulate it and are now included in the revised version of the manuscript.

  1. Table 1 contains the most important information regarding clinical application of CQ/HCQ. The effect of CQ/HCQ on cancer treatment can be moved to a new colume with more detailed information. The patients number (sample size) of clinical trial should be provide

RE 1: We thank the reviewer for her/his suggestion. Table.1 has been edited with more detailed information in the revised version of the manuscript.

  1. The authors raise an important question (there are currently no sensitive and reliable predictive biomarkers for rational selection of cancer patients who could benefit from using CQ/HCQ). Some studies have shown that CQ and HCQ inhibit anti-cancer therapy-induced protective autophagy (e.g., chemotherapy, radiotherapy, and target therapy) to exert anti-cancer effect. The authors may discuss the potential of using autophagy flux as the biomarker for CQ/HCQ

RE 2: We thank the reviewer for her/his suggestion. The potential role of using autophagic flux as a biomarker for CQ/HCQ has been discussed in the revised version of the manuscript.

  1. Please leave a space between the last character and the left parenthesis (e.g., line 38), or between two sentences (e.g., line 60, 79, 91).

RE 3: We thank the reviewer for her/his suggestion which we have implemented in the revised version of the manuscript.

Reviewer 2 Report

Abel-Aziz et al have submitted a manuscript for consideration for publication entitle: “A critical review of chloroquine and hydroxychloroquine as a potential adjuvant agent for treating people with cancer “ for consideration for publication. This is a review of the currently clinical data for treating cancer patients with chloroquine. It gives a details analysis of important factors and criteria needs to be evaluated for chloroquine clinically and results from many clinical trials. Some issues need to be further addressed in the review that were not adequately discussed. See below my comments.

1.       An emphasis was given to the chloroquine ability to inhibit autophagy and rationale why this could be a good strategy to treat cancer. There also needs to be discussion that chloroquine also induces apoptosis independent of autophagy and potential mechanism.

2.       It was mentioned that autophagy can contribute to both cell survival and cell death. This is a double edge sword as depending on the function of autophagy at the time of chloroquine treatment, cancer cells could die or survive especially in the context of combinational treatment. This needs to be further discussed in the manuscript.

3.       In terms of identifying biomarkers for responder subpopulations, there needs to more discussion on this aspect. Many cancer cells have intrinsic drug resistance such as p53 mutation that make them refractory to many treatments. Anti-apoptotic proteins can protect cells as well. How does this play into the effectiveness of chloroquine?

4.       Finally, there are findings that autophagy changes during the course of the development of cancer and microenvironmental factors also alter autopahgy. Hypoxia was discussed but a more fulsome discussion on how this impacts chloroquine treatment would be helpful.

Author Response

Reviewer 2: Abel-Aziz et al have submitted a manuscript for consideration for publication entitle: “A critical review of chloroquine and hydroxychloroquine as a potential adjuvant agent for treating people with cancer “ for consideration for publication. This is a review of the currently clinical data for treating cancer patients with chloroquine. It gives a details analysis of important factors and criteria needs to be evaluated for chloroquine clinically and results from many clinical trials. Some issues need to be further addressed in the review that were not adequately discussed. See below my comments.

RE: We thank the reviewer for her/his interest in our review and the additional points made, that allowed us to better formulate it and are now included in the revised version of the manuscript.

  1. An emphasis was given to the chloroquine ability to inhibit autophagy and rationale why this could be a good strategy to treat cancer. There also needs to be discussion that chloroquine also induces apoptosis independent of autophagy and potential mechanism.

RE 1: We thank the reviewer for her/his suggestion. Autophagy independent mechanisms underlying the anticancer potential of chloroquine have been added in the revised version of the manuscript.

  1. It was mentioned that autophagy can contribute to both cell survival and cell death. This is a double edge sword as depending on the function of autophagy at the time of chloroquine treatment, cancer cells could die or survive especially in the context of combinational treatment. This needs to be further discussed in the manuscript.

RE 2: We thank the reviewer for her/his suggestion. The double-edged role of autophagy has further been discussed in the revised version of the manuscript.

  1. In terms of identifying biomarkers for responder subpopulations, there needs to more discussion on this aspect. Many cancer cells have intrinsic drug resistance such as p53 mutation that make them refractory to many treatments. Anti-apoptotic proteins can protect cells as well. How does this play into the effectiveness of chloroquine?

RE 3: We thank the reviewer for her/his suggestions which we implemented in the revised version of the manuscript.

  1. Finally, there are findings that autophagy changes during the course of the development of cancer and microenvironmental factors also alter autopahgy. Hypoxia was discussed but a more fulsome discussion on how this impacts chloroquine treatment would be helpful.

RE 4: We thank the reviewer for her/his suggestion. The impact of tumor microenvironment on the anticancer potential of chloroquine has been further discussed in the revised version of the manuscript.